# Reporting of Perinatal Outcomes in Probiotic Randomized Controlled Trials. A Systematic Review and Meta-Analysis

**DOI:** 10.3390/nu13010256

**Published:** 2021-01-17

**Authors:** Íñigo María Pérez-Castillo, Rafael Fernández-Castillo, Agustín Lasserrot-Cuadrado, José Luís Gallo-Vallejo, Ana María Rojas-Carvajal, María José Aguilar-Cordero

**Affiliations:** 1Andalusian Research, Development and Innovation Plan, CTS 367, University of Granada, 18001 Granada, Spain; perezcastillo@correo.ugr.es (Í.M.P.-C.); lasserrot@ugr.es (A.L.-C.); Anamar.rojascarvajal@gmail.com (A.M.R.-C.); mariajaguilar@telefonica.net (M.J.A.-C.); 2Faculty of Health Sciences, University of Granada, 18071 Granada, Spain; 3Obstetrics and Gynecology Service, Virgen de las Nieves University Hospital, 18014 Granada, Spain; jgallov@gmail.com

**Keywords:** probiotics, safety, pregnancy, perinatal outcomes, strains, meta-analysis, PRISMA

## Abstract

The use of probiotic microorganisms in clinical practice has increased in recent years and a significant number of pregnant women are regular consumers of these products. However, probiotics might modulate the immune system, and whether or not this modulation is beneficial for perinatal outcomes is unclear. We performed a systematic review and meta-analysis to evaluate the reporting of perinatal outcomes in randomized controlled trials including women supplemented with probiotic microorganisms during pregnancy. We also analyzed the effects that the administration of probiotic microorganisms exerts on perinatal outcomes. In the review, 46 papers were included and 25 were meta-analyzed. Reporting of perinatal outcomes was highly inconsistent across the studies. Only birth weight, cesarean section, and weeks of gestation were reported in more than 50% of the studies. Random effects meta-analysis results showed that the administration of probiotic microorganisms during pregnancy did not have any a positive or negative impact on the perinatal outcomes evaluated. Subgroup analysis results at the strain level were not significantly different from main analysis results. The administration of probiotic microorganisms does not appear to influence perinatal outcomes. Nonetheless, future probiotic studies conducted in pregnant women should report probiotic strains and perinatal outcomes in order to shed light upon probiotics’ effects on pregnancy outcomes.

## 1. Introduction

According to the World Health Organization (WHO), Food and Agriculture Organization (FAO), and the International Scientific Association for Probiotics and Prebiotics (ISAPP), probiotics are defined as “live microorganisms which when administered in adequate amounts confer a health benefit on the host” [1,2]. Similarly, synbiotic products are combinations of probiotic microorganisms and a beneficial substrate constituted by the prebiotic [3]. The use of probiotic microorganisms is widespread, and probiotic/synbiotic products are available in the global market labeled as food supplements or medical products depending on different regulations [4]. The market value of these products was estimated to account for USD 54 billion in 2020 and this trend suggests that it will grow substantially in the next years [5].

Probiotic microorganisms have been used for a plethora of clinical indications such as lactose malabsorption, diarrhea, bowel syndrome, and infection, among others [6]. In pregnant women, the administration of these microorganisms has been proposed to alleviate gastrointestinal symptoms, improve glycemic control, reduce oxidative stress, and lower the incidence of asthma, atopic sensitization, and allergic disease in offspring, among other outcomes [7,8,9]. However, clinical evidence remains far from conclusive [10].

Pregnancy is a dynamic immunological process in which pro- and anti-inflammatory status concur in order to facilitate the different stages of gestation [11]. Whereas a pro-inflammatory stage is necessary for successful embryo implantation and placentation during early pregnancy, an anti-inflammatory switch is required to allow fetal growth during mid-pregnancy. Finally, once fetal development is complete, another physiological pro-inflammatory status leads to labor and delivery [11]. Disruption of this immunological process is linked to adverse perinatal outcomes such as miscarriage, intrauterine growth restriction, and preterm birth [12].

Immune stimulation is among the different mechanisms of action attributed to probiotic microorganisms [13]. One study observed that pregnant women supplemented with a combination of *Bifidobacterium longum*, *Lactobacillus delbrueckii*, and *Streptococcus thermophilus* from 32 weeks of gestation to delivery had higher concentrations of proinflammatory cytokines, namely interleukine-5 (IL-5), interleukine-6 (IL-6), tumor necrosis factor- α (TNF- α), and granulocyte-macrophage colony-stimulating factor (GM-CSF) [14]. On the other hand, a recent study monitored populations of immune cells after the administration of *Lactobacillus reuteri* to pregnant women during mid-gestation, observing that the number of activated regulatory T cells was lower in the group supplemented with *Lactobacillus reuteri* in comparison to the placebo [15]. Hence, the administration of probiotic microorganisms can potentially stimulate or suppress inflammatory status and immune response during gestation, but whether or not these changes are beneficial to pregnancy is unclear [14,16].

Previous reviews have evaluated the effect that the administration of probiotic microorganisms during pregnancy exerts on maternal and perinatal outcomes, concluding that probiotic microorganisms do not increase or decrease the incidence of these outcomes [10,17,18]. These studies have only evaluated these microorganisms at the genus or species level. However, mounting evidence suggests that probiotic effects are strain-dependent, and strain-specificity is usually a poorly reported aspect of probiotics research [19]. Furthermore, studies explicitly designed to assess the safety of probiotic/synbiotic interventions are lacking [20].

In the light of these studies and given the rapid growth of literature regarding probiotic microorganisms, we aimed to evaluate the reporting of perinatal outcomes in randomized controlled trials (RCT) studying the administration of probiotic microorganisms during pregnancy and to explore the associations between the administration of probiotic microorganisms during pregnancy and perinatal outcomes considering the role that strain-specificity could play in the associations.

## 2. Materials and Methods

We conducted a systematic review and meta-analysis of randomized controlled trials studying the administration of probiotic microorganisms during pregnancy. The present systematic review was conducted according to the *Preferred Reporting Items for Systematic Reviews and Meta-Analyses* statement (PRISMA) [21]. The protocol for this study was registered in the PROSPERO database with the number “CRD42020216531.” The PRISMA checklist is presented in Appendix A.

### 2.1. Literature Search

We carried out an automatized search in three databases, PubMed (MEDLINE), Scopus, and Cochrane Controlled Register of Trials (CENTRAL), during August–October 2020. The records dated from inception up to August 2020. The databases were searched by one reviewer. The language was restricted to English, and only full articles published in scientific journals were included in the research. A detailed search strategy based on patients, interventions, comparators, outcomes, and study design (PICOS) chosen for the present study is presented in Appendix A.

### 2.2. Inclusion and Exclusion Criteria

We selected randomized controlled trials enrolling pregnant women in which at least one group of study was treated with probiotic or synbiotic preparations, with specified microorganisms and dosage, and at least one group of comparison received no treatment, routine care, or placebo, independently of the blinding process.

We excluded grey literature such as conference abstracts, workshops, and government reports, as well as other study designs. However, references from systematic reviews were searched for additional articles. Trials studying fermented food or generic products (i.e., probiotic food or probiotic supplement) without specifying the dosage and microorganisms provided were excluded. Studies comparing the use of probiotic preparations against antibiotics, as well as studies without follow-up to delivery, were also excluded. Secondary analyses of previous studies were not included in the review, but they were screened for additional information when not reported in the main analysis and were thus cited in tables when appropriate.

### 2.3. Study Outcomes

The prespecified main outcomes of the study consisted of the number of weeks of gestation, birth weight, preterm birth, cesarean section, low birth weight, macrosomia, small for gestational age (SGA), large for gestational age (LGA), miscarriage, and stillbirth. Secondary outcomes reported in the studies, namely cases of malformation, Apgar test score, umbilical cord pH, anthropometric measures at birth, neonatal death, and admission to neonatal intensive care unit, were also discussed but were not considered for meta-analysis.

Preterm birth was defined as a baby born alive before 37 completed weeks of gestation [22]. SGA was defined as a live birth with a weight below the 10th percentile for the gestation age, while LGA was defined as a live birth with a weight above the 90th percentile [23]. According to ICD-10, low birth weight was considered as a live birth weighing less than 2500 g at delivery [24], while macrosomia was defined as a live birth weighing more than 4000 g at delivery in accordance with the American College of Obstetricians and Gynecologists [25]. When authors provided different definitions for these outcomes, they were specified in tables (Appendix A).

### 2.4. Data Extraction

A template was created for data extraction, and two reviewers independently extracted the data from the included studies. The template consisted of the following items: Authors, year of acceptance for publication, country where the study was conducted, study design, sample size at randomization, sample size of groups analyzed, specific previous conditions (i.e., women with gestational diabetes mellitus), comparison group, main objective of the study, probiotic microorganisms, dosage and posology, vehicle of administration, duration of treatment, conflict of interest, and raw data or statistics on the reported perinatal outcomes (i.e., mean and standard deviation).

Studies using probiotic/synbiotic products provided by a company without stating the manufacturer’s contribution to the paper were considered to present potential conflicts of interest. Disagreements between both reviewers were resolved by a third reviewer.

### 2.5. Quality Assessment

Two reviewers independently evaluated the quality of the included studies using the Cochrane risk-of-bias tool for randomized trials (RoB 2.0) [26]. Five domains were evaluated, including the randomization process, deviations from intended interventions, missing outcome data, measurement of the outcome, and selection of reported results. Follow-up losses >20% were considered as a high risk of bias when assessing missing outcome data. Protocols of included studies were checked when assessing selective report bias. Regarding the overall risk of bias, we applied the following scheme based on the Cochrane Handbook recommendations [27]: One item rated as “high risk of bias” or three or more items rated as “some concerns” = high risk of bias; one or two items rated as “some concerns” = some concerns; all items rated as “low risk of bias” = low risk of bias. Any disagreement was discussed with a third reviewer.

### 2.6. Data Synthesis and Statistical Analysis

We conducted a random effects meta-analysis of the studies included in the review based on criteria selected to avoid potential bias when pooling results. Selected inclusion criteria for the meta-analysis were as follows: (1) Study population consisting of pregnant women without specified previous pathologies, and (2) studies reporting raw data or statistics on the specific perinatal outcome. We did not exclude women at high risk of atopic sensitization from the meta-analysis.

Groups of study assessing additional interventions other than dietary advice (i.e., probiotics + vitamin D) were excluded from analyses. If one study reported two or more groups treated with different probiotic microorganisms, they were combined. Data were pooled as relative risk and 95% confidence interval (95% CI) or mean ± standard deviation (SD), and the I^2^ statistic was used to assess heterogeneity across the studies. We used the Mantel–Haenszel statistical method in all of the analyses. Data reported as mean ± 95% CI were transformed into mean ± SD. These approaches are recommended by the Cochrane handbook [27]. When data were reported as median and interquartile range or median and range, they were transformed into mean ± SD using the method developed by Wan et al. [28]. Additionally, we provided subgroup analyses excluding these estimated statistics in order to assess how these approaches could influence pooled results (Appendix A).

Only outcomes reported in at least two trials that met the aforementioned inclusion criteria were pooled. When preterm birth cases were considered as an exclusion criterium in the papers, we added those preterm birth cases to the groups analyzed. There was only one paper studying synbiotic products that met the inclusion criteria. Therefore, only probiotic studies were finally meta-analyzed.

Subgroup analyses were conducted based on reported strains of microorganisms. Accordingly, we pooled data on outcomes reported in at least two studies using the same microorganism strain or the same combination of strains.

Finally, we created funnel plots for each main analysis including at least 10 studies. Publication bias was assessed by visual inspection of the funnel plots.

All analyses were conducted using Review Manager (RevMan), version 5.4, The Cochrane Collaboration, 2020.

## 3. Results

### 3.1. Selection Process

A total of 26,579 records were screened after removing duplicates, leaving 115 articles available for eligibility, including 10 references from other previous reviews. Of the eligible articles, 87 met the inclusion criteria and were further evaluated to exclude any secondary analysis of a previous research. Finally, 46 RCTs were included in the systematic review and 25 RCTs were included in the meta-analysis. Secondary analyses of previous studies were screened for additional data if they were not reported in the original research, and thus were cited in tables when applicable. The article selection process is presented in Figure 1 according to the PRISMA flow diagram of trials.

### 3.2. Characteristics of the Studies

Characteristics of studies included in the systematic review are presented in Table 1.

In the present review, 46 randomized controlled trials were included. A total of 8519 pregnant women participated in these studies. Of the analyzed studies, 42 trials evaluated the administration of probiotic preparations to pregnant women whereas 4 studies evaluated the administration of synbiotic products. The included papers dated from 2001 to 2020. Of the studies, 15 were conducted in Iran [30,31,33,34,35,36,40,46,47,48,50,51,64,70,72]; 4 trials were conducted in Finland [49,55,56,69]; 3 in New Zealand [66,73,74]; 3 in Australia [37,38,67]; 2 each in Canada [71,75], Germany [43,54], Taiwan [45,68], and Ireland [58,59], and the remaining 13 RCTs were conducted in other different countries. Most of the studies recruited healthy pregnant women. However, 10 trials included only pregnant women with gestational diabetes mellitus [35,36,40,46,48,50,51,52,59,64], 4 studies included exclusively obese/overweight pregnant women [38,44,58,66], and 1 study included only pregnant women carrying group B streptococcus (GBS) [45].

Regarding the main objectives of the studies, the most common were improving insulin/glucose metabolism (13 studies) [30,33,34,40,44,46,50,52,58,64,69,70,72]; preventing eczema, allergic disease, or atopic sensitization (10 studies) [29,37,41,49,53,54,55,65,68,73]; improving oxidative stress status and inflammatory profile (8 studies) [14,36,46,47,48,51,64]; modifying the mother’s or infant’s microbiota (5 studies) [14,61,63,65,75]; and preventing GBS occurrence (4 studies) [45,67,71,74]. Other outcomes included bacterial vaginosis, mastitis, pregnancy outcomes, safety assessment, maternal anthropometric measures, infant colic, genetic profile, and infant diarrhea or gut integrity.

In respect to the microorganisms administered, none of the included studies evaluated the administration of probiotic yeasts (i.e., *Saccharomyces bourlardii*) to pregnant women. In the studies, 21 of the authors administered combinations of *Bifidobacterium* and *Lactobacillus* species [30,32,33,34,35,36,38,41,44,47,48,50,51,52,53,56,60,66,67,69,73]; 18 authors studied the administration of *Lactobacillus* species only [29,31,37,42,43,45,49,54,58,59,63,64,67,68,71,72,74,75]; 4 authors used combinations of *Lactobacillus*, *Bifidobacterium*, and *Streptococcus* species [14,40,46,61]; 1 author evaluated *Bifidobacterium* species only [39]; and the 2 remaining authors evaluated other combinations with different bacterial genera (*Propionibacterium* and *Lactococcus*) [55,65]. Strains of microorganisms administered were reported in 32 of the 46 included studies (70%). The reporting of strains was heterogeneous across the studies, with authors reporting culture collection numbers (i.e., *Lactobacillus rhamnosus* ATCC 53103), commercial designations (i.e., *Bifidobacterium animalis* HN019), or references to the name of the person who originally isolated the strain (i.e., *Lactobacillus rhamnosus* GG). The most frequently used strains, alone or in combination with other strains, were *L. rhamnosus* GG (nine studies) [37,38,41,49,54,55,56,60,68], *B.animalis* BB-12 (seven studies) [33,34,38,40,41,56,66], *L. rhamnosus* GR-1 (six studies) [43,45,63,67,71,75], *L. reuteri* RC-14 [43,45,63,67,71,75], *L. acidophilus* LA-5 (four studies) [33,34,40,41], and *L. rhamnosus* HN001 (three studies) [69,73,74].

Probiotic/synbiotic administration was only compared to placebo in 37 trials (80%) [29,30,31,32,36,37,38,39,40,41,42,43,44,45,46,47,49,50,51,52,53,54,55,58,59,63,64,65,66,68,70,71,72,73,74,75], while 5 studies (11%) used a group with no treatment or routine care as a comparison [14,33,34,35,67], and the remaining 4 papers (9%) analyzed other additional comparison groups (i.e., probiotic + vitamin D) [48,56,60,69]. Microorganisms were administered orally in all of the included studies. Administration vehicles consisted of capsules in most of the cases (35 studies, 76%), with 3 studies using probiotic yoghurt (6.5%) [33,34,70]; 3 studies using probiotic powder [53,60,61]; and 1 study each using probiotic oil (2.2%) [29], tablets [14], and milk [41]. The two remaining studies (4.3%) did not specify the vehicle used to administer the microorganisms [67,72].

The dosages of individual microorganisms were diverse, ranging from 5 × 10^5^ CFU [14] to 5 × 10^10^ CFU [64]. However, in some cases, the posology was unclear [34,35]. In other cases, the dosages were defined as CFU/g, but the authors did not declare the mass of the product administered.

Mean treatment duration was approximately 9 weeks, ranging from 3 weeks of duration [67] to 26 weeks [56]. Women were treated with probiotic/synbiotic preparations during the third trimester of gestation in most of the trials (34 studies, 74%) [2,3,4,5,6,7,8,9,10,11,12,13,14,15,16,17,18,19,20,21,22,23,24,25,26,27,28,29,30,31,32,33,34,35,36,37,39,40,41,42,45,46,48,49,52,53,54,55,58,59,60,61,64,65,67,68,70,71,72,73], through second and third trimesters in 7 studies (15%) [38,44,50,56,66,74,75], through first and second trimesters in 3 studies [43,47,69], and independently of the trimester of gestation in 1 study [63]. One author did not specify the period of treatment [51].

Quality assessment results showed that 18 studies (39%) had a low risk of bias (high methodological quality) [29,30,31,34,35,36,37,38,43,44,52,55,56,58,59,66,69,73], 15 studies (33%) had an unclear risk of bias due to concerns in one or two of the items assessed [41,47,49,50,51,52,53,54,60,64,65,68,70,71,73,74], and the remaining 13 studies (28%) had a high risk of bias (low methodological quality) [14,32,33,39,40,42,45,46,48,61,63,67,75]. Detailed results of the risk of bias assessment are presented in Appendix A. Regarding conflicts of interest, 9 studies (20%) showed no conflict of interest (clearly declaring manufacturer contribution to the manuscript) [38,41,44,58,59,61,66,69,74], 28 studies (60%) had a potential conflict of interest (product supplied by a private laboratory and unclear manufacturer contribution to the manuscript) [14,30,31,33,34,35,36,37,40,42,43,45,46,47,48,49,50,51,52,56,63,64,67,68,70,71,72,75], and the other 9 studies had an existing conflict of interest (either declared conflict of interest or clear existing relationship with the manufacturer) [29,32,39,53,54,55,60,65,73].

### 3.3. Reporting of Perinatal Outcomes

A summary of the studies included in the systematic review and in the meta-analysis based on the reporting of perinatal outcomes is presented in Table 2. Detailed data on the perinatal outcomes reported in the included studies are presented in Appendix A. Finally, cases of miscarriage and stillbirth were not meta-analyzed due to a lack of consensus on the outcome definition across the studies.

As presented in Appendix A, the reporting of perinatal outcomes was completely heterogeneous across the studies. Only two authors described, in detail, the perinatal outcomes in the cohort of study reporting data on all of the following items: Cases of miscarriage/stillbirth, weeks of gestation, cases of preterm birth, birth weight, cases of macrosomia or LGA, cases of low birth weight or SGA, and cesarean section [38,69]. On the other hand, eight studies did not report data on any of the aforementioned items [30,33,35,40,42,47,67,72]. Regarding the main outcomes of the present review, the most frequently reported outcome across the studies was birth weight (29 studies, 63%), followed by cesarean section (27 studies, 59%), weeks of gestation (24 studies, 52%), preterm birth (<37 weeks of gestation, 21 studies, 46%), miscarriage/stillbirth (11 studies, 24%), macrosomia (10 studies, 22%), LGA (7 studies, 15%), SGA (5 studies, 11%), and low birth weight (1 study, 2%). Other outcomes reported in the studies consisted of neonatal hospitalization or admission to neonatal intensive care unit (NICU) (15 studies, 33%) [29,34,36,37,38,45,48,51,58,59,66,69,70,71,74], anthropometric measures at birth (i.e., head circumference or birth length) (15 studies) [14,29,34,36,48,49,50,51,56,59,60,66,70,73,74], Apgar test score at 1 or 5 minutes (11 studies, 24%) [14,45,48,51,56,59,60,69,71,74,75], malformation/fetal abnormalities (7 studies, 15%) [34,38,44,55,58,59,69], hypoglycemia (6 studies, 13%) [36,38,48,51,52,69], hyperbilirubinemia or jaundice (5 studies, 11%) [34,36,38,48,51], induction of labor (3 studies, 7%) [44,59,75], polyhydramnios (3 studies) [36,48,51], cord blood pH (2 studies, 4%) [69,75], preterm birth (<35 weeks of gestation, 2 studies) [46,64], preterm birth (<34 weeks of gestation, 1 study, 2%) [38], and umbilical artery resistance (1 study) [39].

### 3.4. Administration of Probiotics during Pregnancy and Length of Gestation

Eleven studies evaluated the administration of probiotic preparations to pregnant women without previous pathologies reporting the cases of preterm delivery (<37 weeks of gestation) in the groups analyzed. The random effects meta-analysis of these studies showed that probiotic administration during pregnancy did not have a statistically significant impact on the odds of preterm birth in the groups of study (RR = 1.19, 95% CI = 0.81–1.74). Heterogeneity across the studies was low (I^2^ = 14%; Figure 2). Visual inspection of the funnel plot did not reveal potential publication bias (Figure 3).

We conducted a subgroup analysis excluding the trials that provided the probiotics from the last 4–6 weeks of gestation to avoid bias due to the timing of the intervention. The subgroup analysis results were similar to those obtained in the main analysis (RR = 1.03, 95% CI = 0.53–1.99, I^2^ = 31%; Appendix A).

Among included studies, three authors evaluated the administration of *Lactobacillus rhamnosus* GR-1 in combination with *Lactobacillus reuteri* RC-14 and provided data on the odds of PTB. The subgroup analysis of these studies did not show a significant association between the administration of these strains during pregnancy and preterm birth (RR = 0.76, 95% CI = 0.17–3.35, I^2^ = 34%; Appendix A).

Thirteen trials reported the length of gestation, providing the number of weeks of gestation in both intervention and comparison groups. Data were reported as mean ± standard deviation in eight of the studies, while the remaining five studies reported weeks of gestation as mean + 95% CI or median + range or interquartile range.

The meta-analysis of the 13 studies showed that the administration of probiotics was not associated with the number of weeks of gestation (MD = 0.03, 95% CI = −0.21–0.27). However, heterogeneity across the studies was considerable (I^2^ = 78%) (Figure 4). We did not observe publication bias from visual inspection of the funnel plot (Figure 5).

We conducted a subgroup analysis of the eight studies that provided data on the weeks of gestation as mean ± standard excluding the estimated values from the other five studies, yielding similar results (MD = 0.09, 95% CI = −0.14–0.32, I^2^ = 56%; (Appendix A).

Three of these studies specifically evaluated the administration of *Lactobacillus rhamnosus* GG. The subgroup analysis of these studies did not show significant changes in the pooled mean difference of the weeks of gestation (MD = −0.38, 95% CI = −0.92–0.15, I^2^ = 56%; Appendix A). Similarly, the subgroup analysis of the two studies that used a combination of *Lactobacillus rhamnosus* GR-1 and *Lactobacillus reuteri* RC-14 did not significantly influence the meta-analysis results (MD = 0.18, 95% CI = −0.75–1.11, I^2^ = 86%; Appendix A).

### 3.5. Administration of Probiotics during Pregnancy and Birth Weight

Thirteen studies reported data on birth weight either as mean ± standard deviation (eight studies) or as mean + 95% CI or median + range/interquartile range (five studies). The random effects meta-analysis of these studies showed that administration of probiotics during pregnancy did not have a significant impact on birth weight (MD = −5.36, 95% CI = −37.60–26.89) (Figure 6). Heterogeneity across included studies was low (I^2^ = 0%). We did not observe publication bias after visual inspection of funnel plots (Figure 7).

Subgroup analysis of only the eight studies that provided birth weight as mean ± standard deviation did not significantly modify the meta-analysis results (MD = 10.80, 95% CI = −28.03–49.62, I^2^ = 0%; Appendix A). In the same fashion, subgroup analysis excluding studies conducted during the last 4-6 weeks of gestation did not modified the meta-analysis results (MD = −7.43, 95% CI = −80.27–65.41.01, I^2^ = 18%; Appendix A).

We also separately analyzed two studies that used *Lactobacillus rhamnosus* GG providing data on birth weight yielding similar results (MD = 16.80, 95% CI = −82.40–116.01, I^2^ = 0%; Appendix A).

None of the studies that provided the cases of low birth weight (<2500 g at birth) or SGA (birth weight <10th percentile for the gestational age) met the inclusion criteria for the meta-analysis. On the other hand, three studies reported the cases of macrosomia (>4000 g at birth) in the cohort of study and two other trials reported the cases of LGA (birth weight >90th percentile for the gestational age). However, Sahhaf et al. [70] did not specify the definition of macrosomia used. The meta-analysis of these studies showed that probiotic consumption during pregnancy was not associated with an increased risk of macrosomia or LGA (macrosomia: RR = 0.84, 95% CI = 0.30–2.34, I^2^ = 60%; LGA: RR = 0.98, 95% CI = 0.60–1.61, I^2^ = 0%; Figure 8 and Figure 9). Exclusion of the study conducted by Sahhaf et al. [70] (which was the main cause of heterogeneity) did not significantly modify this correlation (data not shown).

### 3.6. Administration of Probiotics during Pregnancy and Cesarean Section

Seventeen studies reported the cases of cesarean section in the groups of study and met the inclusion criteria for the meta-analysis. The random effects meta-analysis of these studies showed that administration of probiotics during pregnancy did not influence cesarean section rate (RR = 0.98, 95% CI = 0.87–1.10; Figure 10). The studies were homogeneous (I^2^ = 0%). Visual inspection of the funnel plot did not reveal publication bias (Figure 11).

Two authors evaluated the administration of *L. rhamnosus* GG and provided data on cesarean section cases. The subgroup analysis of these studies did not substantially modify the meta-analysis results (RR = 0.85, 95% CI = 0.52–1.38, I^2^ = 0%; Appendix A). In line with this, the subgroup analysis of two other authors that administered a combination of *L. rhamnosus* GR-1 and *L. reuteri* RC-14 did not show a significant correlation with cesarean section (RR = 0.85, 95% CI = 0.46–1.59, I^2^ = 0%; Appendix A).

## 4. Discussion

The present systematic review and meta-analysis aimed at evaluating the reporting of perinatal outcomes in randomized controlled trials analyzing the effect that the administration of probiotic microorganisms might exert on these outcomes. We included 46 studies in the present review, 25 of which were meta-analyzed. The random effects meta-analysis results showed that the administration of probiotics during pregnancy was not associated with any perinatal outcome evaluated. However, the reporting of perinatal outcomes was completely heterogeneous across the included studies, and only birth weight, cases of cesarean section, and weeks of gestation were reported in more than 50% of the studies.

By definition, probiotic microorganisms, when administered in adequate amounts, confer a health benefit on the host [1,2]. Nonetheless, probiotics must also be safe for their intended use [76]. In this sense, an exhaustive review on this subject including 384 RCTs involving probiotics, prebiotics, and synbiotics concluded that the reporting of adverse events in most of these trials was lacking and inadequate [77]. Pregnant women are an immunologically vulnerable population, and the mechanisms by which probiotic microorganisms might stimulate or suppress the immune system are not clear [76]. Therefore, not only adverse events (which is out of the scope of the present review), but also perinatal outcomes should be considered when conducting probiotic trials in pregnant women.

Sample size is usually a limitation of probiotic trials when assessing differences in perinatal outcomes [78,79]. In this regard, meta-analyses help to overcome this limitation by pooling data from several studies. In line with the present research, other reviews and meta-analyses on this topic have concluded that probiotics and synbiotics do not positively or negatively influence perinatal outcomes [10,17,18]. Nonetheless, the authors of these reviews coincide on affirming that the reporting of these outcomes is highly heterogeneous, with some authors reporting only weeks of gestation at delivery [68] or cases of preterm labor [64] without further considering other outcomes. Furthermore, some authors have reported perinatal outcomes in secondary analyses even when they were not provided in the original research [57,62]. This could imply selective report bias when assessing probiotics safety.

The meta-analysis results showed that the administration of probiotics was not associated with preterm birth (RR = 1.19, 95% CI = 0.81–1.74) or birth weight (MD = −5.36, 95% CI = −37.60–26.89). The lack of association could be caused by the timing of the intervention given that several included studies administered probiotic preparations from the last 4–6 weeks of gestation to delivery or postpartum. However, the subgroup analysis excluding these studies did not yield different results. Therefore, it seems unlikely that the timing of the intervention would have a significant impact on meta-analysis conclusions. Nonetheless, it is remarkable that only 142 cases of preterm birth were included in the meta-analysis from a total of 2934 participants, which implies a low rate of preterm delivery in these studies analyzed (4.8%) compared to global estimates (10.6%) [80]. Similar rates of preterm birth were observed in another meta-analysis [10]. Given the low reported incidence of preterm birth in the studies analyzed, future studies with a bigger sample size are required to evaluate the effects of probiotics on this outcome.

Birth weight was mostly reported as a continuous variable in the studies. In this sense, SGA and low birth weight were clearly underreported outcomes across the studies and could not be included in the meta-analysis. On the other hand, the majority of the studies reporting cases of LGA or macrosomia were conducted in pregnant women with GDM and were thus excluded from the analyses. Finally, only three studies reporting cases of macrosomia and two studies reporting LGA cases were meta-analyzed. Due to the low number of pooled studies, it is not possible to draw solid conclusions from the meta-analyses regarding LGA and macrosomia. More studies are necessary to evaluate the influence of probiotics administration on SGA and low birth weight.

Mounting evidence suggests that probiotics’ effects are strain-dependent [19]. However, the meta-analyses conducted to date were not able to pool data from specific strains due to the heterogeneity of included studies. In the present review, we conducted subgroup analyses of studies evaluating the administration of *L. rhamnosus* GG, *L. rhamnosus* GR-1, and *L. reuteri* RC-14. In this regard, *L. rhamnosus* GR-1 has been suggested to present beneficial properties for the prevention of preterm birth in animal models [81]. Our results showed that the administration of *L. rhamnosus* GG was not associated with higher or lower birth weight in the included studies. Similarly, neither *L. rhamnosus* GG nor *L. rhamnosus* GR-1 in combination with *L. reuteri* RC-14, were associated with the number of weeks of gestation of cesarean section rate. However, almost one-third of the authors did not report the probiotic strains administered, and the small number of studies included in the sub-analyses makes it impossible to draw strong conclusions from these results.

Finally, the present research is not exempt from limitations. We did not exclude studies analyzing twin pregnancies, which could bias our results given that multiple pregnancies have a higher risk of adverse perinatal outcomes. We did not ask the authors of the studies for nonreported data, which could have increased the number of studies meta-analyzed. Lastly, we did not analyze maternal outcomes (i.e., preeclampsia) or adverse events (i.e., diarrhea), which are subjects of much deeper research on probiotic safety.

## 5. Conclusions

The meta-analysis results at the genus or species level showed that the administration of probiotic microorganisms during pregnancy does not have any positive or negative impact on birth weight, length of gestation, and cesarean section. The sub-analyses at the strain level did not modify these results. However, the number of studies reporting the same perinatal outcome and evaluating the administration of the same probiotic strains was very low, and it is thus not possible to draw strong conclusions from the sub-analyses results regarding the effect that specific probiotic strains might exert on perinatal outcomes. The reporting of perinatal outcomes was inconsistent across the included studies. Future probiotic randomized controlled trials should report perinatal outcomes and probiotic strains in order to shed light on the effects that probiotic microorganisms exert on pregnancy, placing emphasis on the safety of these interventions.

## Figures and Tables

**Figure 1 nutrients-13-00256-f001:**
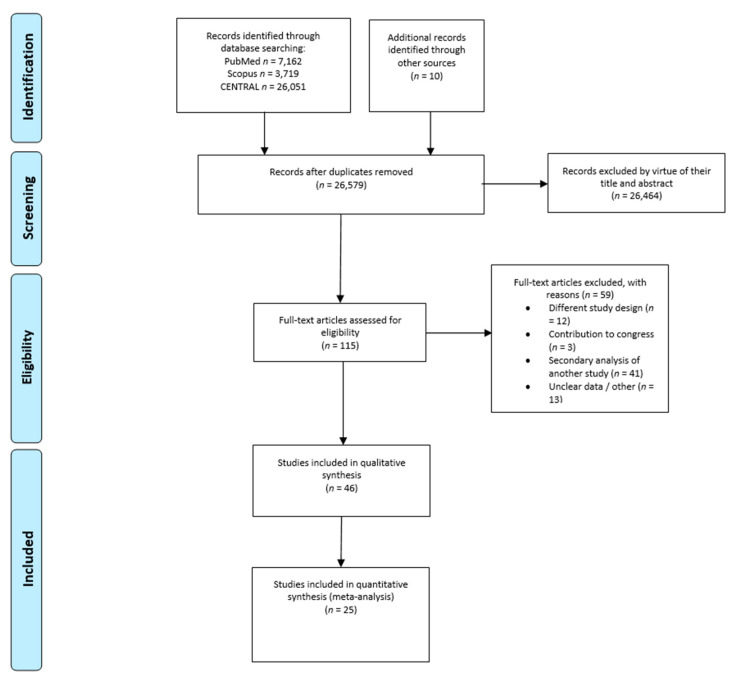
Preferred Reporting Items for Systematic Reviews and Meta-Analysis (PRISMA) flow diagram of trials.

**Figure 2 nutrients-13-00256-f002:**
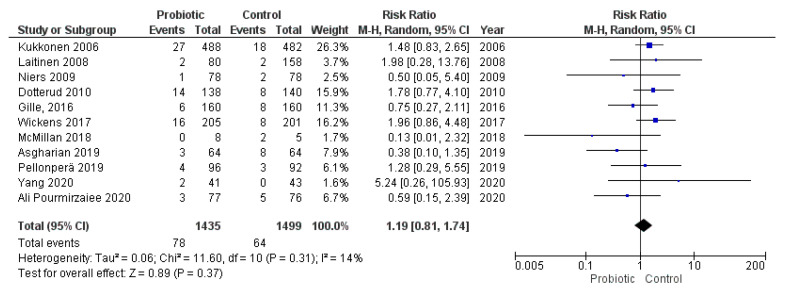
Meta-analysis of studies evaluating the administration of probiotic and preterm birth.

**Figure 3 nutrients-13-00256-f003:**
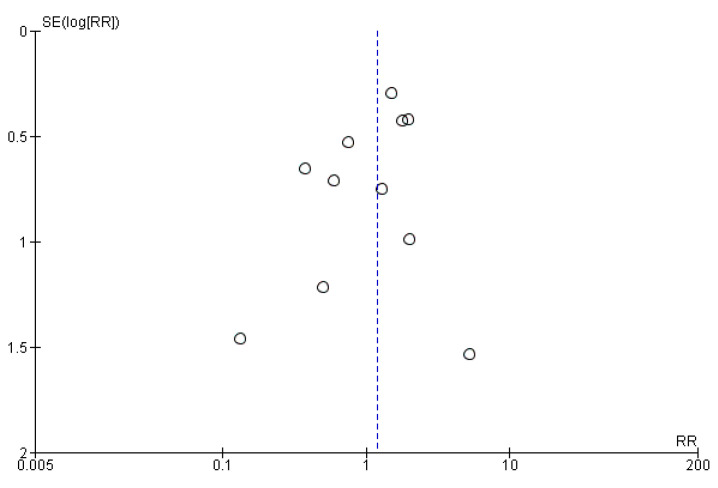
Funnel plot of studies evaluating the administration of probiotic and preterm birth.

**Figure 4 nutrients-13-00256-f004:**
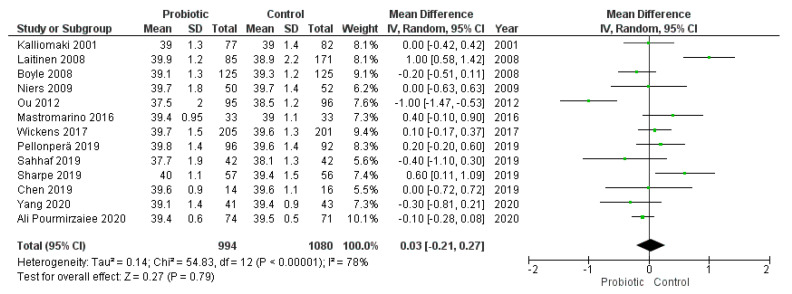
Meta-analysis of studies evaluating the administration of probiotic and weeks of gestation.

**Figure 5 nutrients-13-00256-f005:**
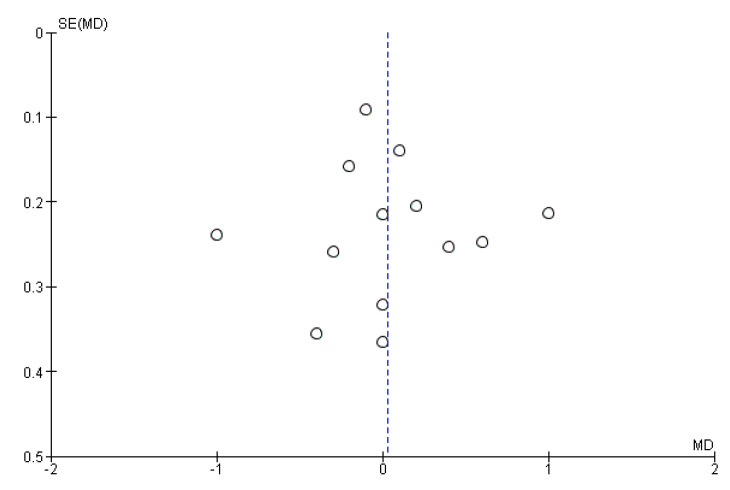
Funnel plot of studies evaluating the administration of probiotic and weeks of gestation.

**Figure 6 nutrients-13-00256-f006:**
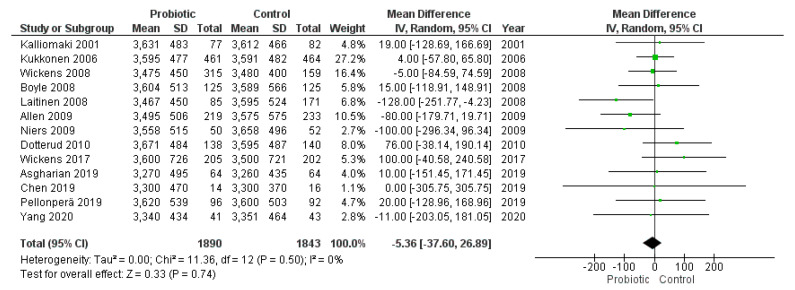
Meta-analysis of studies evaluating the administration of probiotic and birth weight.

**Figure 7 nutrients-13-00256-f007:**
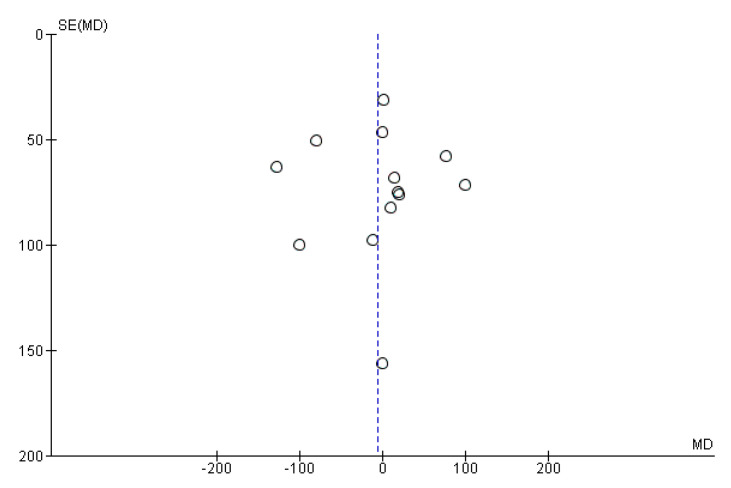
Funnel plot of studies evaluating the administration of probiotic and birth weight.

**Figure 8 nutrients-13-00256-f008:**
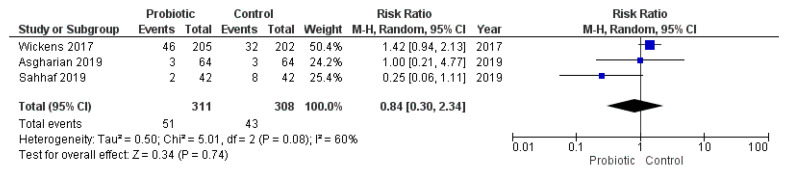
Meta-analysis of studies evaluating the administration of probiotic and macrosomia.

**Figure 9 nutrients-13-00256-f009:**
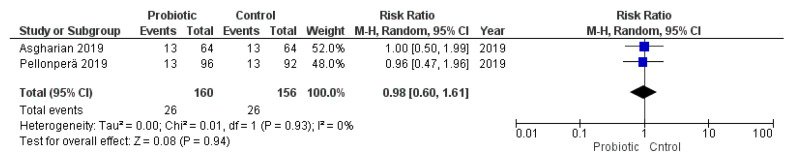
Meta-analysis of studies evaluating the administration of probiotic and large for gestational age.

**Figure 10 nutrients-13-00256-f010:**
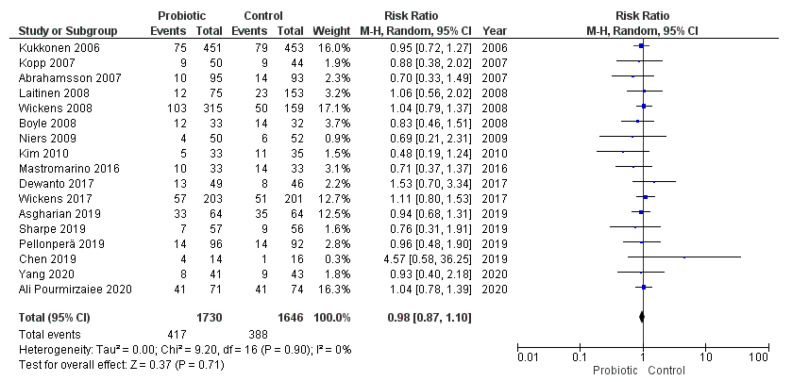
Meta-analysis of studies evaluating the administration of probiotic and cesarean section.

**Figure 11 nutrients-13-00256-f011:**
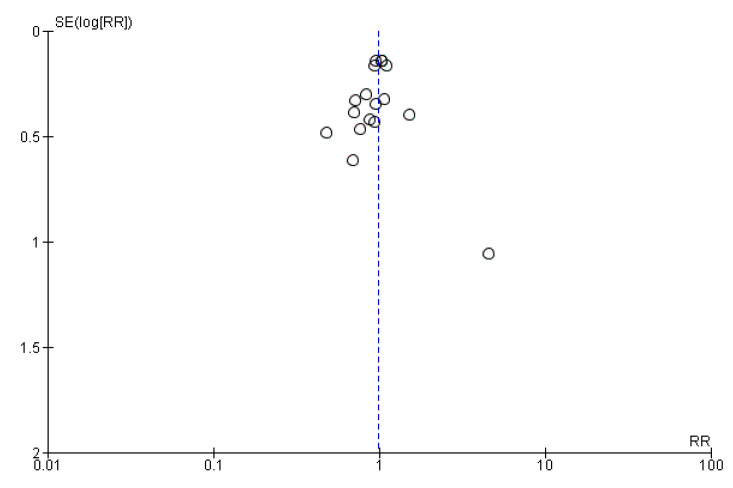
Funnel plot of studies evaluating the administration of probiotic and cesarean section.

**Table 1 nutrients-13-00256-t001:** Characteristics of the included studies.

Author/Country/Year of Acceptance for Publication/Trial Registration	Sample Size at Randomization	Groups Analyzed at Delivery	Microorganism (Strain)/Dosage (Posology) + Additional Interventions	Administration Vehicle	Treatment Duration	Main Clinical Outcome	Risk of Bias Assessment	Conflict of Interest
Abrahamsson et al./Sweden/2007/Not reported [29]	232	Probiotic = 95 Placebo = 93	*Lactobacillus**reuteri* (ATCC 55730)/1 × 10^8^ CFU (daily)	Oil	From 36 WoG to delivery	Eczema and sensitization in offspring	Low	Existing conflict of interest
Ahmadi et al./ Iran/2016/IRCT201605085623N77 [30]	70	Synbiotic = 35 Placebo = 35	*Lactobacillus acidophilus*/2 × 10^8^ CFU *Lactobacillus casei*/2 × 10^8^ CFU *Bifidobacterium bifidum*/2 × 10^8^ CFU + Inulin/0.8 g (daily)	Oral capsule	Six weeks from 24–28 WoG	Insulin metabolism and lipid profile	Low	Potential conflict of interest
Ali Pourmirzaiee et al./ Iran/2020/IRCT201311004014882N7 [31]	175	Probiotic = 74 Placebo = 71	*Lactobacillus reuteri* (LR92, DSM 26866)/1 × 10^8^ CFU (daily)	Oral capsule	Four last weeks of gestation	Prevention of infantile colic	Low	Potential conflict of interest
Allen et al./United Kingdom/2009/ISRCTN26287422 [32]	454	Probiotic = 220 Placebo = 234	*Lactobacillus salivarius* (CUL61, NCIMB 30211)/6.25 × 10^9^ CFU *Lactobacillus paracasei* (CUL08, NCIMB 30154) /1.25 × 10^9^ CFU *Bifidobacterium animalis* (CUL34, NCIMB 30172)/1.25 × 10^9^ CFU *Bifidobacterium bifidum* (CUL20, NCIMB 30153)/1.25 × 10^9^ CFU (daily)	Oral capsule	From last month of gestation to postpartum	Prevalence of adverse events	High	Existing conflict of interest
Asemi et al./Iran/2011/Not reported [33]	82	Probiotic = 37 Control = 33	*Lactobacillus acidophilus* (LA-5)/1 × 10^7^ CFU *Bifidobacterium animalis* (Bb-12)/1 × 10^7^ CFU (200 g daily)	Yoghurt	Nine weeks from the third trimester of gestation	Insulin resistance and insulin levels	High	Potential conflict of interest
Asgharian et al./ Iran/2019/IRCT201604013706N31 [34]	130	Probiotic = 64 Control = 64	*Lactobacillus acidophilus* (LA-5)/5 × 10^8^ CFU/g *Bifidobacterium animalis* (Bb-12)/5 × 10^8^ CFU/g (100 g daily)	Yoghurt	From 24 WoG to delivery	Glucose levels	Low	Potential conflict of interest
Bababi et al./ Iran/2008/IRCT20171010036697N1 [35]	50 pregnant women with GDM	Probiotic = 24 Control = 24	*Lactobacillus acidophilus*/2 × 10^9^ CFU/g *Lactobacillus casei*/2 × 10^9^ CFU/g *Bifidobacterium bidifum*/2 × 10^9^ CFU/g *Lactobacillus fermentum*/2 × 10^9^ CFU/g (posology not clearly defined)	Oral capsule	Six weeks from 24–28 WoG	Genetic and metabolic profile	Low	Potential conflict of interest
Badehnoosh et al./ Iran/2017/IRCT201611115623N91 [36]	60 pregnant women with GDM	Probiotic = 30 Placebo = 30	*Lactobacillus acidophilus*/2 × 10^9^ CFU/g *Lactobacillus casei*/2 × 10^9^ CFU/g *Bifidobacterium bidifum*/2 × 10^9^ CFU/g (Posology not clearly defined)	Oral capsule	Six weeks from 24–28 WoG	Oxidative stress and inflammation biomarkers	Low	Potential conflict of interest
Boyle et al./Australia/2008/Cochrane Skin Group Trial No. 36 [37]	250	Probiotic = 125 Placebo = 125	*Lactobacillus rhamnosus* (GG)/1.8 × 10^10^ CFU (daily)	Oral capsule	From 36 WoG to delivery	Risk of eczema during infancy	Low	Potential conflict of interest
Callaway et al./Australia/2018/ACTRN12611001208998 [38]	433 overweight/obese pregnant women	Probiotic = 207 Placebo = 204	*Lactobacillus rhamnosus* (GG)/1 × 10^9^ CFU *Bifidobacterium animalis* (Bb-12)/1 × 10^9^ CFU (daily)	Oral capsule	From 16 WoG to delivery	Prevalence of GDM	Low	No conflict of interest
Chen et al./China/2019/Not reported [14]	32	Probiotic = 14 Control = 16	*Bifidobacterium longum*/5 × 10^6^ CFU *Lactobacillus delbrueckii bulgaricus*/5 × 10^5^ CFU *Streptococcus thermophilus*/5 × 10^5^ CFU (two tablets twice a day)	Oral tablet	From 32 WoG to delivery	Gut microbiota and pro-inflammatory cytokine profile	High	Potential conflict of interest
Dewanto et al./Indonesia/2017/Not reported [39]	110	Probiotic = 36 Placebo = 38	*Bifidobacterium animalis* (HNO 19)/1 × 10^9^ CFU (daily)	Oral capsule	From third trimester to postpartum	Infant’s gut mucosal integrity	High	Existing conflict of interest
Dolatkhan et al./ Iran/2015/IRCT201405181597N3 [40]	64 pregnant women with GDM	Probiotic = 27 Placebo = 29	*Lactobacillus acidophilus* (LA-5)/4 × 10^9^ CFU *Bifidobacterium animalis* (Bb-12)/4 × 10^9^ CFU *Streptococcus thermophilus* (STY-31)/4 × 10^9^ CFU *Lactobacillus delbrueckii bulgaricus* (LBTY-27)/4 × 10^9^ CFU (daily)	Oral capsule	Eight weeks from 24–28 WoG	Glucose metabolism and gestational weight gain	High	Potential conflict of interest
Dotterud et al./Norway/2010/NCT00159523 [41]	415	Probiotic = 138 Placebo = 140	*Lactobacillus rhamnosus* (GG)/5 × 10^10^ CFU *Bifidobacterium animalis* (Bb-12)/5 × 10^10^ CFU *Lactobacillus acidophilus* (LA-5) /5 × 10^9^ CFU (daily)	Milk	From 36 WoG to postpartum	Atopic sensitization and allergic disease in offspring	Some Concerns	No conflict of interest
Fernández et al./ Spain/2015/NCT01505361 [42]	108	Probiotic = 55 Placebo = 53	*Lactobacillus salivarius* (PS2)/1 × 10^9^ CFU (daily)	Oral capsule	From 27–32 WoG to delivery	Mastitis	High	Potential conflict of interest
Gille et al./Germany/2016/ISRCTN40042090 [43]	320	Probiotic = 160 Placebo = 160	*Lactobacillus rhamnosus* (GR-1)/1 × 10^9^ CFU *Lactobacillus reuteri* (RC-14)/1 × 10^9^ CFU (daily)	Oral capsule	Eight weeks from <12 WoG	Bacterial vaginosis	Low	Potential conflict of interest
Halkjær et al./Denmark/2020/NCT02508844 [44]	50 obese pregnant women	Probiotic = 20 Placebo = 23	*Bifidobacterium breve* (DSM 24,732) *Bifidobacterium longum* (DSM 24,736) *Bifidobacterium infantis* (DSM 24,737) *Lactobacillus delbrueckii bulgaricus* (DSM 24,734)/ 1.12 × 10^9^ CFU total (twice a day)	Oral capsule	14–20 WoG–delivery	Glucose homeostasis and gestational weight gain	Low	No conflict of interest
Ho et al./Taiwan/2015/NCT01577108 [45]	110 GBS (+) pregnant women	Probiotic = 49 Placebo = 50	*Lactobacillus rhamnosus* (GR-1)/1 × 10^9^ CFU *Lactobacillus reuteri* (RC-14)/1 × 10^9^ CFU (twice a day)	Oral capsule	From 35–37 WoG to delivery	Occurrence of streptococcus GBS	High	Potential conflict of interest
Jafarnejad et al./Iran/2016/Not reported [46]	82 pregnant women with GDM	Probiotic = 41 Placebo = 41	*Streptococcus thermophilus* *Bifidobacterium* *Breve* *Bifidobacterium longum* *Bifidobacterium infantis* *Lactobacillus* *Acidophilus* *Lactobacillus plantarum* *Lactobacillus* *Paracasei* *Lactobacillus delbrueckii Bulgaricus*/112.5 × 10^9^ CFU total (twice a day)	Oral capsule	Eight weeks from 26 WoG (mean)	Glycemic control and inflammatory status	High	Potential conflict of interest
Jamilian et al./ Iran/2016/IRCT201503035623N38 [47]	60	Probiotic = 30 Placebo = 30	*Lactobacillus acidophilus*/2 × 10^9^ CFU *Lactobacillus casei*/2 × 10^9^ CFU *Bifidobcterium bifidum*/2 × 10^9^ CFU (daily)	Oral capsule	12 weeks from 9 WoG	Metabolic profile, inflammatory factors and oxidative stress	Some Concerns	Potential conflict of interest
Jamilian et al./ Iran/2018/IRCT201706075623N119 [48]	90 pregnant women with GDM	Probiotic = 29 Probiotic + Vitamin D = 30 Placebo = 28	*Lactobacillus acidophilus*/2 × 10^9^ CFU *Bifidobacterium bifidum*/2 × 10^9^ CFU *Lactobacillus reuteri*/2 × 10^9^ CFU *Lactobacillus fermentum*/2 × 10^9^ CFU (daily)	Oral capsule	Six weeks from 24–28 WoG	Metabolic profile, inflammatory factors and oxidative stress	High	Potential conflict of interest
Kalliomaki et al./Finland/2001/Not reported [49]	159	Probiotic = 77 Placebo = 82	*Lactobacillus rhamnosus* (GG)/1 × 10^10^ CFU (twice a day)	Oral capsule	From 2–4 weeks before delivery to postpartum.	Atopic sensitization	Some Concerns	Potential conflict of interest
Karamali et al./ Iran/2016/IRCT201601035623N63 [50]	60 pregnant women with GDM	Probiotic = 30 Placebo = 30	*Lactobacillus acidophilus*/2 × 10^9^ CFU *Lactobacillus casei*/2 × 10^9^ CFU *Bifidobcterium bifidum*/2 × 10^9^ CFU (daily)	Oral capsule	Six weeks from 24–28 WoG	Glycemic control and lipid profile	Some Concerns	Potential conflict of interest
Karamali et al./ Iran/2017/IRCT201704205623N108 [51]	60 pregnant women with GDM	Synbiotic = 30 Placebo = 30	*Lactobacillus acidophilus (T16, IBRC-M10785)*/2 × 10^9^ CFU *Lactobacillus casei* (T2, IBRC-M10783)/2 × 10^9^ CFU *Bifidobacterium bifidum* (T1, IBRC-M10771)/2 × 10^9^ CFU + inulin/800 mg (daily)	Oral capsule	Six weeks of duration (Commencement of treatment not specified)	Inflammation and oxidative stress	Some Concerns	Potential conflict of interest
Kijmanawat et al./Thailand/2018/Thai Clinical Trials Registry Number 20170606002 [52]	60 pregnant women with GDM	Probiotic = 28 Placebo = 29	*Lactobacillus acidophilus*/1 × 10^9^ CFU *Bifidobacterium bifidum*/1 × 10^9^ CFU (daily)	Oral capsule	Four weeks from 24–28 WoG	Insulin resistance	Low	Potential conflict of interest
Kim et al./Korea/2009/ISRCTN26134979 [53]	112	Probiotic = 33 Placebo = 35	*Bifidobacterium bifidum* (BGN4)/1.6 × 10^9^ CFU *Bifidobacterium animalis* (AD011)/1.6 × 10^9^ CFU *Lactobacillus acidophilus* (AD031)/1.6 × 10^9^ CFU (daily)	Powder	From 4–8 weeks before delivery to postpartum	Eczema in offspring	Some Concerns	Existing conflict of interest
Kopp et al./Germany/2007/UKF000505 [54]	105	Probiotic = 50 Placebo = 44	*Lactobacillus rhamnosus* (GG, ATC 53013)/5 × 10^9^ CFU (twice a day)	Oral capsule	From 4–6 weeks before delivery to postpartum	Atopic disease in offspring	Some Concerns	Existing conflict of interest
Kukkonen et al./Finland/2006/Not reported [55]	1223	Probiotic = 461 Placebo = 464	*Lactobacillus rhamnosus* (GG, ATC 53103)/5 × 10^9^ CFU *Lactbacillus rhamnosus* (LCT705, DSM 7061)/5 × 10^9^ CFU *Bifidobacterium breve* (Bb99, DSM 13692)/2 × 10^8^ CFU *Propionibacterium freudenreichii shermanii* (JS, DSM 7076)/2 × 10^9^ CFU (Twice a day)	Oral capsule	From 36 WoG to delivery	Allergic disease in offspring	Low	Existing conflict of interest
Laitinen et al./Finland/2008/NCT00167700 [56,57]	256	Diet + Probiotic = 85 Diet + Placebo = 86 Control + Placebo = 85	*Lactobacillus rhamnosus* (GG, ATCC 53103)/1 × 10^10^ CFU *Bifidobacterium animalis* (Bb-12)/1 × 10^10^ CFU + Dietary counseling (daily)	Oral capsule	From 14 WoG to postpartum	Infant’s metabolic status and maternal anthropometric measures	Low	Potential conflict of interest
Lindsay et al./Ireland/2014/ISRCTN97241163(A) [58]	175 obese pregnant women	Probiotic = 63 Placebo = 75	*Lactobacillus salivarius* (UCC118)/1 × 10^9^ CFU (daily)	Oral capsule	From 24 to 28 WoG	Glucose status	Low	No conflict of interest
Lindsay et al./Ireland/2015/ISRCTN97241163(B) [59]	149 pregnant women with GDM	Probiotic = 74 Placebo = 75	*Lactobacillus salivarius* (UCC118)/1 × 10^9^ CFU (daily)	Oral capsule	From <34 to delivery	Metabolic parameters and pregnancy outcomes	Low	No conflict of interest
Mantaring et al./Philippines/2018/NCT01073033 [60]	233	Probiotic + food supplement = 60 Food supplement = 62 Control = 61	*Bifiboacterium animalis* (CNCC I-3446)/7 × 10^8^ CFU *Lactobacillus rhamnosus* (CGMCC I-3724)/7 × 10^8^ CFU (twice a day)	Powder	From 24–28 WoG to postpartum	Infant’s diarrhea	Some Concerns	Existing conflict of interest
Mastromarino et al./Italy/2015/NCT01367470 [61,62]	67	Probiotic = 33 Placebo = 33	*Lactobacillus paracasei* (DSM 24733) *Lactobacillus plantarum* (DSM 24730) *Lactobacillus acidophilus* (DSM 24735) *Lactobacillus delbrueckii bulgaricus* (DSM 24734) *Bifidobacterium longum* (DSM 24736) *Bifidobacterium breve* (DSM 24732) *Bifidobacterium infantis* (DSM 24737) *Streptococcus thermophilus* (DSM 24731)/9 × 10^9^ CFU total (daily)	Powder	From 36 WoG to postpartum	Breast milk bacteria	High	No conflict of interest
McMillan et al./ Rwanda/2018/NCT02150655 [63]	38	Probiotic = 8 Placebo = 5	*Lactobacillus rhamnosus* (GR-1)/1 × 10^9^ CFU *Lactobacillus reuteri* (RC-14)/1 × 10^9^ CFU (daily)	Oral capsule	One month from 4–32 WoG	Vaginal microbiota	High	Potential conflict of interest
Nabhani et al./ Iran/2018/IRCT201511183140N16 [64]	95 pregnant women with GDM	Synbiotic = 45 Placebo = 45	*Lactobacillus acidophilus*/5 × 10^10^ CFU/g *Lactobacillus plantarum*/1.5 × 10^10^ CFU/g *Lactobacillus fermentum*/7 × 10^9^ CFU/g *Lactobacillus gasseri*/2 × 10^10^ CFU/g + FOS/38.5 mg (500 mg daily)	Oral capsule	Six weeks from 24–28 WoG	Insulin resistance, lipid profile and antioxidative status	Some Concerns	Potential conflict of interest
Niers et al./Netherlands/2009/NCT00200954 [65]	156	Probiotic = 50 Placebo = 52	*Bifidobacterium bifidum* (W23)/1 × 10^9^ CFU *Bifidobacterium animalis* (W52)/1 × 10^9^ CFU *Lactococcus lactis* (W58)/1 × 10^9^ CFU (daily)	Oral capsule	From the last six weeks of pregnancy to postpartum	Eczema in offspring, microbial colonization and immune response	Some Concerns	Existing conflict of interest
Okense-Gafa et al./New Zealand/2018/ACTRN12615000400561 [66]	230 obese pregnant women	Probiotic = 115 Placebo = 115	*Lactobacillus rhamnosus* (GG)/7 × 10^9^ CFU *Bifidobacterium animalis* (Bb-12)/7 × 10^9^ CFU (daily)	Oral capsule	From 12–17 WoG to delivery	Gestational weight gain and pregnancy outcomes	Low	No conflict of interest
Olsen et al./Australia/2017/Not reported [67]	34	Probiotic = 7 Control = 13	*Lactobacillus rhamnosus* (GR-1) *Lactobacillus reuteri* (RC-14)/1 × 10^8^ CFU total (daily)	Not specified	Three weeks from 36 WoG	Occurrence of streptococcus GBS	High	Potential conflict of interest
Ou et al./Taiwan/2012/IDNCT00325273 [68]	191	Probiotic = 95 Placebo = 96	*Lactobacillus rhamnosus* (GG, ATCC 53103)/1 × 10^10^ CFU (daily)	Oral capsule	From 24 WoG to postpartum	Allergic disease	Some Concerns	Potential conflict of interest
Pellonperä et./Finland/2019/NCT01922791 [69]	439	Probiotic + Placebo = 110 Fish oil + Probiotic = 109 Fish oil + Placebo = 109 Placebo + Placebo = 110	*Lactobacillus rhamnosus* (HN001, ATCC SD5675)/1 × 10^10^ CFU *Bifidobacterium animalis* (420, DSM 22089)/1 × 10^10^ CFU (daily)	Oral capsule	From <18 WoG to postpartum	Risk of GDM and glucose metabolism	Low	No conflict of interest
Sahhaf et al./ Iran/2019/IRCT20121224011862N2 [70]	84	Probiotic = 42 Placebo = 42	*Lactobacillus acidophilus* *Bifidobacterium animalis* /1 × 10^6^ CFU total (300 mg daily)	Yoghurt	Eight weeks from 24–28 WoG	Glycemic parameters	Some Concerns	Potential conflict of interest
Sharpe et al./Canada/2019/NCT02528981 [71]	139	Probiotic = 57 Placebo = 56	*Lactobacillus rhamnosus* (GR-1)/2.5 × 10^9^ CFU *Lactobacillus reuteri* (RC-14)/2.5 × 10^9^ CFU (twice a day)	Oral capsule	Twelve weeks from 23–25 WoG	Occurrence of streptococcus GBS	Some Concerns	Potential conflict of interest
Taghizadeh et al./ Iran/2013/IRCT201212105623N3 [72]	56	Synbiotic = 26 Placebo = 26	*Lactobacillus sporogenes*/9 × 10^7^ CFU + Inulin/0.36 g (twice a day)	Not specified	Nine weeks from the third trimester of gestation	Glycemic status and C-reactive protein sensitivity	Some Concerns	Potential conflict of interest
Wickens et al./New Zealand/2008/ACTRN12607000518460 [73]	512	Probiotic 1 = 157 Probiotic 2 = 158 Placebo = 159	Probiotic 1 = *Lactobacillus rhamnosus* (HN001)/6 × 10^9^ CFU Probiotic 2 =*Bifidobacterium animalis animalis* (HN019)/9 × 10^9^ CFU (daily)	Oral capsule	From 35 WoG to postpartum	Eczema and atopic sensitization in offspring	Low	Existing conflict of interest
Wickens et al./New Zealand/2017/ACTRN12612000196842 [74]	423	Probiotic = 206 Placebo = 202	*Lactobacillus* *rhamnosus* (HN001)/6 × 10^9^ CFU (daily)	Oral capsule	From 14–16 WoG to postpartum	Occurrence of streptococcus GBS	Some Concerns	No conflict of interest
Yang et al./Canada/2020/NCT01697683 [75]	86	Probiotic = 41 Placebo = 43	*Lactobacillus rhamnosus* (GR-1)/2.5 × 10^9^ CFU *Lactobacillus reuteri* (RC-14)/2.5 × 10^9^ CFU (daily)	Oral capsule	Twelve weeks from 12–16 WoG	Vaginal microbiota, chemokines and cytokines profile	High	Potential conflict of interest

CFU: Colony-Forming Unit; GBS: Group B Streptococcus; GDM: Gestational Diabetes Mellitus; FOS: Fructooligosaccharide. WoG: Week of Gestation.

**Table 2 nutrients-13-00256-t002:** Summary of included studies bases on the reporting of perinatal outcomes.

Systematic Review				Meta-Analysis			
Reported Perinatal Outcome	Number of Reviewed Studies	*n*	Intervention	Number of Pooled Studies	*n*	Pooled RR/MD (95% CI)	I^2^
Preterm Birth	19	4903	Probiotic	11	2934	RR = 1.16 (0.78–1.71)	16%
	2	155	Synbiotic	0	-	-	-
Weeks of Gestation	23	4144	Probiotic	13	2074	MD = 0.03 (−0.21–0.27)	78%
	1	60	Synbiotic	0	-	-	-
Birth Weight	28	6666	Probiotic	13	3578	MD = −5.57 (−38.48–27.34)	0%
	1	60	Synbiotic	0	-	-	-
Low Birth Weight	1	433	Probiotic	0	-	-	-
	0	-	Synbiotic	0	-	-	-
Macrosomia	10	1654	Probiotic	3	94	RR = 0.84 (0.30–2.34)	60%
	1	60	Synbiotic	0	-	-	-
SGA	5	1301	Probiotic	0	-	-	-
	0	-	Synbiotic	0	-	-	-
LGA	7	1436	Probiotic	2	316	RR = 0.98 (0.60–1.61)	0%
	0	-	Synbiotic	0	-	-	-
Cesarean Section	26	5952	Probiotic	17	3445	RR = 0.93 (0.83–1.04)	0%
	1	60	Synbiotic	0	-	-	-
Miscarriage or Stillbirth	11	2595	Probiotic	0	-	-	-
	0	-	Synbiotic	0	-	-	-

LGA: Large for Gestational Age; MD: Mean Difference; RR: Relative Risk; SGA: Small for Gestational Age.

## Data Availability

Data sharing not applicable.

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
