# Peer review of "Reporting of Perinatal Outcomes in Probiotic Randomized Controlled Trials. A Systematic Review and Meta-Analysis"

_nutrients, 2021, doi:10.3390/nu13010256_

Round 1

Reviewer 1 Report

This systematic review and meta-analysis focused on a topic that has gained a lot of attention, lately, that of prenatal probiotic supplementation and the effects on the infant. The topic is interesting, and the performed analysis is described in detail.

The authors should comment on the following:

As the authors mentioned probiotic action is strain-specific (lines 70,71 and 418). Therefore, in my opinion, the conclusion that they reached in lines 437-439 (‘administration of probiotic microorganisms during pregnancy does not have any positive or negative impact on….’), could be misleading. Moreover, although the authors examined the effects of L. rhamnosus GG and of L. rhamnosus GR-1 and L. reuteri RC-14 in subgroup analysis, and concluded that results at the strain level were not significantly different from main analysis results, a very small number of studies were included in each case. This issue is very prevalent in probiotic research in general, as not many clinical studies are conducted with the same probiotic strain against a specific condition or disease.

Author Response

Reviewer #1 comments

Comments and Suggestions for Authors

This systematic review and meta-analysis focused on a topic that has gained a lot of attention, lately, that of prenatal probiotic supplementation and the effects on the infant. The topic is interesting, and the performed analysis is described in detail.

The authors should comment on the following:

1          As the authors mentioned probiotic action is strain-specific (lines 70,71 and 418). Therefore, in my opinion, the conclusion that they reached in lines 437-439 (‘administration of probiotic microorganisms during pregnancy does not have any positive or negative impact on….’), could be misleading. Moreover, although the authors examined the effects of L. rhamnosus GG and of L. rhamnosus GR-1 and L. reuteri RC-14 in subgroup analysis, and concluded that results at the strain level were not significantly different from main analysis results, a very small number of studies were included in each case. This issue is very prevalent in probiotic research in general, as not many clinical studies are conducted with the same probiotic strain against a specific condition or disease.

1          We are grateful to receive your comments in order to improve our paper. Regarding conclusions, we agree that the statement made in lines 437-439 could be misleading given that we did not specify if these conclusions make reference to analyses at the strain or the species level.  To solve this issue, we modified lines 437-438 as follows: Meta-analysis results at the genus or species level showed that the administration of probiotic microorganisms during pregnancy does not have any positive or negative impact on birth weight, length of gestation, and cesarean section. We also agree that the conclusions drawn from the sub-analysis results at the strain level are not strong given the small number of included studies. Therefore, we modified lines 440 and 443 as follows: Sub-analyses at the strain level did not modify these results. However, the number of studies reporting the same perinatal outcome and evaluating the administration of the same probiotic strains was very low and it is thus not possible to draw strong conclusions from sub-analysis results regarding the effect that specific probiotic strains might exert on perinatal outcomes. We consider that these changes provide better support for the conclusions stated in lines 444-446 “Future probiotic randomized controlled trials should report perinatal outcomes and probiotic strains in order to shed light upon the effects that probiotic microorganisms exert on pregnancy, placing emphasis on the safety of these interventions”. Thank you.

Reviewer 2 Report

The manuscript “Reporting of Perinatal Outcomes in Probiotic Randomized Controlled Trials. A Systematic Review and Meta-analysis” reports the null effect of the administration of probiotic microorganisms during pregnancy at least on the perinatal outcomes evaluated.

In my view, this is a very important and interesting topic which might have a great impact on pregnant women and even on the health of the newborns. I agree with the authors, the huge heterogeneity in the current research in the field impairs the potential knowledge and conclusions extracted from them.

I only have minor comments to the authors:

Numbers are mostly written in digits but sometimes authors use letters (eg. ten instead of 10) Please, unify the criteria.

Authors tend to start sentences with numbers (eg. line 174 number 87). Please change to letters.

Define RCT.

Reference 19: doi is in blue and underlined.

Author Response

Reviewer’s #2 comments

Comments and Suggestions for Authors

The manuscript “Reporting of Perinatal Outcomes in Probiotic Randomized Controlled Trials. A Systematic Review and Meta-analysis” reports the null effect of the administration of probiotic microorganisms during pregnancy at least on the perinatal outcomes evaluated.

1          In my view, this is a very important and interesting topic which might have a great impact on pregnant women and even on the health of the newborns. I agree with the authors, the huge heterogeneity in the current research in the field impairs the potential knowledge and conclusions extracted from them.

1          We appreciate the reviewer’s comments on our manuscript in order to improve its quality and the feedback received. Thank you.

I only have minor comments to the authors:

2          Numbers are mostly written in digits but sometimes authors use letters (eg. ten instead of 10) Please, unify the criteria.

3          Authors tend to start sentences with numbers (eg. line 174 number 87). Please change to letters.

2 – 3     Thank you for pointing out these mistakes. After thoughtfully revising the manuscript, we conducted the following changes:

-           Line 21: 46 to Forty-six

-           Line 113: the 10th percentile

-           Line 167: ten studies to 10 studies

-           Line 173: 26,579 records to A total of 26,579 records

-           Line 174: ten references to 10 references… 87 articles to Eighty-seven articles.                             

-           Table 1: Sharpe et al. and Yang et al. 12 weeks to Twelve weeks

-           Line 187: 46 Randomized controlled trials to Forty-six randomized controlled trials

-           Line 188: 42 to Forty-two

-           Line 190: 15 studies to Fifteen studies (one space was deleted in line 190)

-           Line 195: ten trials to 10 trials

-           Line 201: ten studies to 10 studies

-           Line 208: 21 authors to Twenty-one authors

-           Line 283: 11 studies to Eleven studies

-           Line 303: 13 trials to Thirteen trials

-           Line 325: 13 studies to Thirteen studies

-           Line 358: 17 studies to Seventeen studies

4          Define RCT.

4          We defined RCT the first time the term appears in the main text. Therefore, we changed lines 75-76 from: we aimed to evaluate the reporting of perinatal outcomes in randomized controlled trials studying the administration of probiotic microorganisms… to we aimed to evaluate the reporting of perinatal outcomes in randomized controlled trials (RCT) studying the administration of probiotic microorganisms…

5          Reference 19: doi is in blue and underlined.

5          We corrected this mistake and checked the other references. Thank you.

Round 2

Reviewer 1 Report

The revised manuscript is suitable for publication